# Experimental and Numerical Investigation of Ballistic Resistance of Polyurea-Coated Aluminum Plates under Projectile Impacts

Chenglong He [1,2], Yaqing Liu [2,*], Yingkang Yao [1,*] and Qihui Chen [2]

1   Hubei Key Laboratory of Blasting Engineering of Jianghan University, Wuhan 430056, China; hechenglong@nuc.edu.cn
2   Shanxi Province Key Laboratory of Functional Nanocomposites, North University of China, Taiyuan 030051, China; cqh@nuc.edu.cn
*   Correspondence: lyq@nuc.edu.cn (Y.L.); shanxiyao@jhun.edu.cn (Y.Y.)

**Abstract:** The effects of the spraying thickness and the position on the response of aluminum plates under impact loading were studied. The impact tests and numerical simulation were conducted for the penetration process of polyurea-coated 2024 aluminum plates with tungsten sphere impacts. The results indicate the impact resistance performance is similar at slower impact velocity (500–1000 m/s), and the front (or double-side) coating has a smaller advantage. When the impact velocity rises to 1500 m/s, the back coating has a better energy-absorbing performance. The polyurea perform more efficiently with the increase in the impact velocity because the elastomer has large-scale deformation. By comparing the different thicknesses of the back coating, the residual velocity of the fragment has small changes and the impact energy absorption increased with the increase in the coating thickness. The separated phenomenon is serious in front of the bonding face with shear compression failure. In the back polyurea layer, the stripping area is smaller than the front bonding face, and the petaloid cracking is formed with tensile failure.

**Keywords:** spray polyurea; ballistic resistance; damage mechanics; impact behavior

## 1. Introduction

Owing to the standard spraying process technology promotion and the excellent properties of polyurea materials, aluminum plates with polyurea layers are becoming more prevalent in marine applications. These military structures not only withstand the impact and corrosion of seawater in everyday practice, but they are also subject to explosion shock waves and high-speed projectile impacts in wartime environments [1–3]. The dynamic responses of polyurea-coated aluminum plates are directly affected by the dynamic load process. Therefore, it is important to explore the failure mechanism of composite plate for different impact conditions, especially the quantitative evaluation of the effect of polyurea coatings on the response of plates under projectile impact.

Polyurea is a protective coating for a metal plate that absorbs part of the fragment impact energy because polyurea material has intrinsic elastomeric properties under dynamic loading [4–6]. The failure mode and the dynamic response process of polyurea material are mainly controlled by the loading conditions because polyurea is a strain rate sensitive material, and its strength increases with the loading rates for tensile or puncture loads [7–9]. The performance of polymer-coated metal laminates is greatly affected by loading conditions [10,11]. The polymer layer suppresses plugging and diffuses plastic deformation in the metal layer, and it has a significant effect on the response of the steel plate under dynamic loads [12–14]. Liu et al. [15] explored the ballistic resistance of carbon fiber reinforced plastics and polyurea-coated composites. The polyurea layer changed the evolution of the metal necking process and the polyurea delayed the necking onset

phenomenon [16]. Ackland et al. [17] found that polyurea coatings exhibited a debonding phenomenon within a circular region. Roland et al. [18] and Gamache et al. [19] conducted a comprehensive investigation on the ballistic penetration-resistant properties of elastomer-metal laminate armor. Jiang et al. [20] found the maximum displacements were reduced when the tank surface was coated with polyurea.

The relative position of the polyurea layer influences the dynamic performance of steel plates under dynamic loading [21,22]. The polyurea-improved steel plate has blast resistance when it is coated on the back face. However, it promotes the likelihood of failure of a steel plate if polyurea is placed on the front face [23,24]. When the polyurea is located on the front side of the steel plate, the high-hardness polyurea exhibits superior performance compared to a low-hardness polyurea [25]. The resistance capability of polyurea coated on steel plates nonlinearly increase when the thickness increases [26,27]. Mohotti et al. [28,29] discussed the ability of a polyurea coating to act as a protective layer, and established ballistic limit curves. Sharma et al. [30] determined the impact response of 15 mm thick AA2014-T652 plates at 800–1300 m/s in impacting. Mostofi et al. [31] explored the residual deformations of polyurea-coated aluminum plates.

In the numerical simulation analysis of the blast resistance of the polyurea-reinforced panels, Mohotti et al. [32] investigated the dynamic response of polyurea coating along steel plates under blasting by using LS-DYNA. Dewapriya and Miller [33,34] simulated the penetration performance of multilayer polyurea/silicon-carbide under ballistic impact by using molecular dynamics. Liu et al. [35] researched the dynamic response of polyurea plates and self-closing behavior during penetration by using AUTODYN.

Our study refers to an experimental and numerical evaluation of the polymer thickness and relative position influence on the penetration performance of 2024 aluminum-polyurea composite plates subjected to fragment impacts (300–1400 m/s). In this ballistic experimental study, several metallic specimens were coated with 2, 4, 6 mm (back side) and 1 mm (double sides) thickness polyurea layers. Both high-speed photography and digital image correlation methods were adopted to calculate the strain field, and X-ray computed tomography (CT) was used to show the microscopic damage and interfacial debonding. In addition, the simulated target stress and kinetic energy attenuation were compared to the experimental data.

## 2. Experimental and Simulation Setup

### 2.1. Experimental Setup

Experiments were conducted at the Laboratory of Explosion at the Beijing Institute of Technology. As shown in Figure 1, expanding gases were produced that drove the sabot and projectile movement along the gun barrel after the propellants were ignited. The baffle plate was set up to prevent the sabot from moving between the muzzle and the speed test plate. The velocities of the projectile were measured by the speed test plate and an NDG202G-2 electronic tester. The specimen was fixed to the frame using clamps at all four edges, and the specimen was situated in a fixed position during impact. DIC as a non-contact measurement was used to obtain the strain field and the deformation process. An iX Cameras™ High-Speed Camera i-SPEED 508 was used with Nikon lenses, and the high-strength glass was set up to protect the cameras. The images of the specimen were captured by high-speed cameras with 15,000 frames per second (fps).

The area of interest (AOI) size was 200 × 200 mm with approximately 320 × 320 pixels (as seen in Figure 2). The speckle pattern should be randomly distributed with an appropriate density, good contrast, and each speckle should be at least three pixels in size [36,37]. In this experiment, the diameter of the speckle was 2 mm, and a 200 × 200 mm zone was marked on the spray-painted surface. The Von Mises strain fields were calculated using Match ID software, with a subset of 17 and a step of 6. When calculating the strain field by DIC analysis, two side areas with less light are dismissed and the analysis region is selected as 150 mm × 200 mm to reduce calculation error.

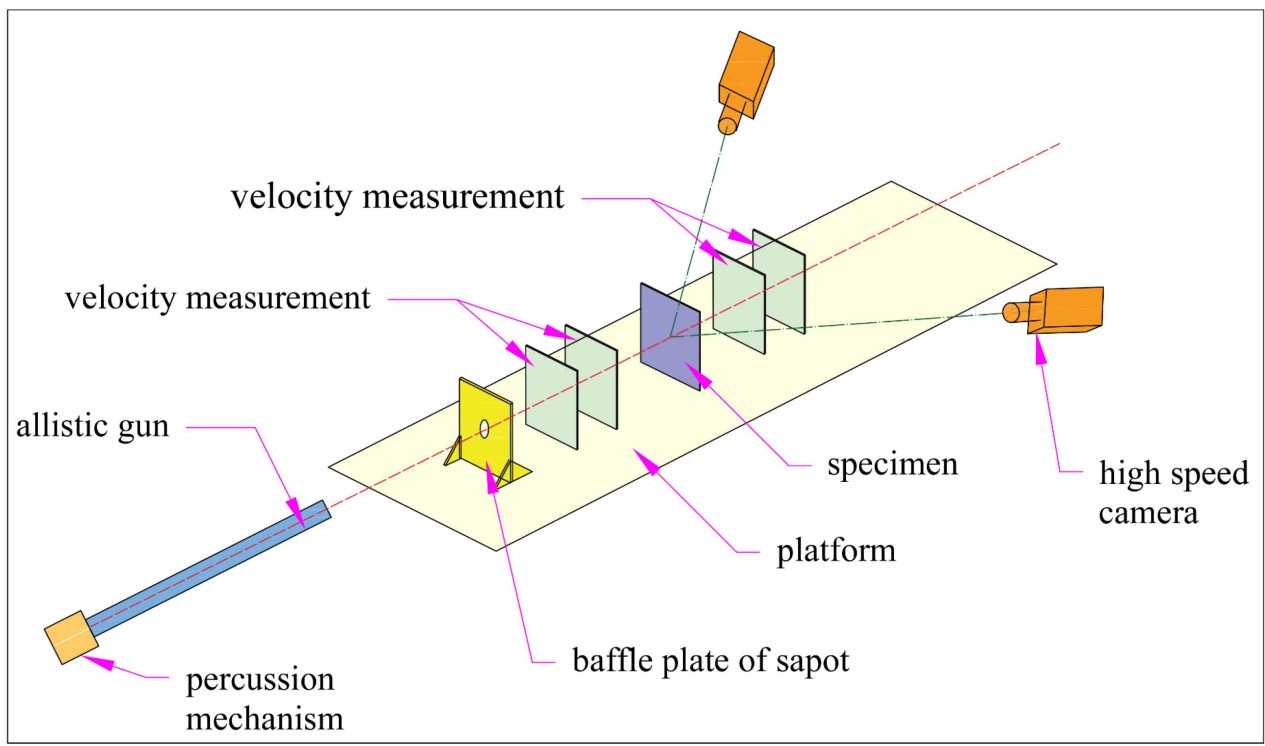

**Figure 1.** A schematic view of the experimental setup.

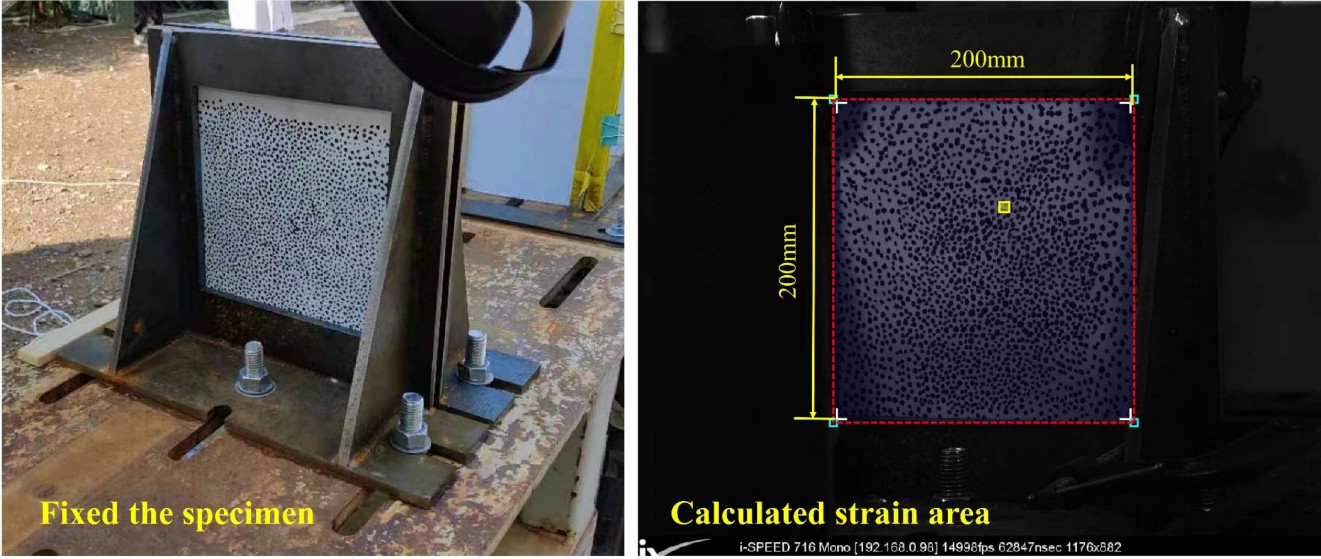

**Figure 2.** AOI zones and schematic diagram of the experiment.

Five experiments were carried out in this study. Tests 1–3 were designed to study the relative coating relative position effects on the response process of the composite plate. Tests 3–5 were designed to investigate the effects of the spraying thickness on the penetration resistance process with impacts at different speeds, as shown in Table 1. The configurations "1P/10Al/1P" in the table represent the 1 mm polyurea layers that were placed on double sides of the 10 mm aluminum plate.

**Table 1.** Schematic representation of different plate configurations.

| Group | Configuration | Geometry | Plate Thickness (mm) | Areal Densities (kg/m$^2$) | Impact Velocity (m/s) |
|---|---|---|---|---|---|
| 1 | 10Al | | 10 | 27.3 | 500/1000/1400 |
| 2 | 1P/10Al/1P | | 1 + 10 + 1 | 27.5 | 500/1000/1400 |
| 3 | 10Al/2P | | 10 + 2 | 27.5 | 500/1000/1400 |
| 4 | 10Al/4P | | 10 + 4 | 27.7 | 500/1000/1400 |
| 5 | 10Al/6P | | 10 + 6 | 27.9 | 500/1000/1400 |

■ 2024 Aluminium plates　　■ Polyurea coating

*2.2. Materials*

A gun barrel with a diameter of 14.5 mm was used to fire the projectile against the target plates. The tungsten alloy material was selected as an equivalent material for the blast shells and projectiles because of its high strength and hardness. In the impact tests, the projectiles were made into spheres with 8 mm diameters and 4.7 g weights, and the initial velocities of the projectile were in the range of 500–1400 m/s with the adjustment of the quantity of the propellants. The projectiles were supported by foam in a cylindrical polycarbonate sabot as they traveled down a gun barrel. The sabot also prevented the gaseous explosive energy from escaping and helped the projectile to achieve high-speed flight.

In the experiment, 2024 aluminum plates were selected because of their anisotropic characteristics with regard to initial yield strength and plastic deformation. The aluminum plates were made with 250 × 250 × 10 mm thicknesses in the Shanghai Hechuan Metal Material Co., Ltd. (Shanghai, China). The quasi-static properties of the 2024 aluminum material were tested. The tensile and compression failure stresses were 410 MPa and

684 MPa. The properties parameters were as follows: average density = 2.790 g/cm$^3$, Poisson's ratio = 0.33, and Young's modulus = 73 GPa.

The Air++1608 polyurea was supplied from the Qingdao Air++ New Materials Co., Ltd. (Qingdao, China), and it was mainly composed of Semi-prepolymer, modified amino polyether, and amine chain extender. The physical properties were obtained and more mechanical properties are shown in Table 2.

**Table 2.** Mechanical properties of polyurea in a quasi-static state.

| Parameters | Unit | Value |
|---|---|---|
| Young's modulus | MPa | 51.2 |
| Tangent Modulus | MPa | 1.9 |
| Tensile strength | MPa | 18.0 |
| Density | g/cm$^3$ | 1.01 |
| Poisson's Ratio | | 0.4 |

To improve adhesion between the aluminum and the coating layer, cleaning, derusting, and sandblasting surface treatment were essential for the aluminum plate. Special spraying equipment and a spray gun were used during the spraying process, and it was ensured that the coating had a uniform particle distribution, compact structure, and controllable thickness. The polyurea coating was solidified at seven days at 23 ± 2 °C with a relative humidity of 50 ± 5% after spraying. The main performance indicators of polyurea coating and mechanical properties are shown in Table 3.

**Table 3.** Parameters of polyurea coating.

| Parameters | Unit | Value |
|---|---|---|
| Gelation time | s | 10 |
| Tensile strength | MPa | 18 |
| Tear strength | N/mm | 80 |
| Adhesion strength (concrete) | MPa | 3.5 |
| Adhesion strength (steel) | MPa | 11 |
| Hardness | | 85–95 |
| Oil resistance | | No rust, no foaming, no shedding |
| Resistant to liquid media | | No rust, no foaming, no shedding |

*2.3. Simulation Model*

Numerical analyses are a useful tool in dynamic research [38]. ABAQUS/Explicit, as a finite element model, has been widely used in solving the penetration process. Within this work, the model structure was built based on the experiment, and the model length was reduced to 120 mm for higher-quality mesh modeling. The "Hard Contact Pair" contact algorithm was selected in the interactions between the fragment and the coated plate. The element size of the impact zone was 0.5 mm, and the number of composite plate elements was approximately 600,000. The cohesive elements were inserted between the aluminum and the polymer layers to estimate the interfacial separation process. The total time was 100 μs, and the key points that were located in the middle of the projectile were selected to analyze the velocity decay process, as shown in Figure 3.

The Johnson-Cook model is popular for modeling the behavior of metal materials subjected to dynamic loading, and it is used for investigating impact problems. In the simulation, the Johnson-Cook model was adopted for both the tungsten ball and the aluminum plate; the parameters are shown in Table 4.

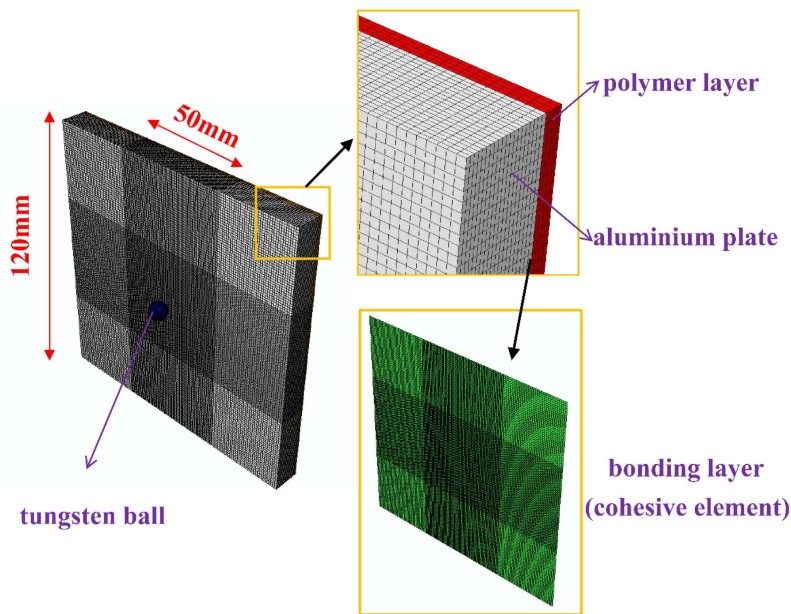

**Figure 3.** Geometrical models and local features in simulation.

**Table 4.** Parameters of Johnson-Cook model.

| Parameters | | Unit | 2024 Aluminum | Tungsten |
|---|---|---|---|---|
| **Mechanical properties** | | | | |
| Density | $\rho_0$ | (g/cm$^3$) | 2.78 | 18 |
| Young's modulus | $E$ | (GPa) | 73 | 4100 |
| Poisson's ratio | $v$ | | 0.33 | 0.3 |
| **Johnson-Cook constitutive model** | | | | |
| Initial yield stress | $A$ | (MPa) | 369 | 1300 |
| Hardening constant | $B$ | (MPa) | 684 | 0 |
| Strain rate constant | $C$ | | 0.0083 | 0.06 |
| Hardening exponent | $n$ | | 0.73 | 1 |
| Thermal softening exponent | $m$ | | 1.7 | 0 |
| Room temperature | $T_r$ | K | 294 | 294 |
| Melting temperature | $T_m$ | K | 772 | 17,900 |
| Reference strain rate/s | | | 1 | |
| **Johnson-Cook damage model** | | | | |
| Damage constant | $D_1$ | | 0.13 | |
| Damage constant | $D_2$ | | 0.13 | |
| Damage constant | $D_3$ | | −1.5 | |
| Damage constant | $D_4$ | | 0.011 | |
| Damage constant | $D_5$ | | 0 | |
| Displacement at failure | | | 0.7 | |

A large number of literature references have shown that polyurea materials have a strain rate effect as a viscoelastic material, and the yield strength increases with the strain rate increase. In this research, the multistage elastoplastic material model was adopted to analyze the response of the polyurea coating during impact. The material parameters were obtained from different stress-strain curves of polyurea at different strain rates in the literature [7]. In this experiment, the dynamic tensile strength of polyurea rise up to 27.38 MPa and failure strain is 0.069 when the strain rate is 490 (1/s); more material parameters are listed in Table 5.

**Table 5.** Material parameters of stress-strain curve in simulation.

| Stress (MPa) | Strain | Strain Rate (1/s) | Stress (MPa) | Strain | Strain Rate (1/s) | Stress (MPa) | Strain | Strain Rate (1/s) |
|---|---|---|---|---|---|---|---|---|
| 0 | 0 | 0 | 0 | 0 | 90 | 0 | 0 | 490 |
| 1 | 0.2 | 0 | 0.5 | 0.1 | 90 | 12 | 0.1 | 490 |
| 5 | 0.1 | 0 | 12 | 0.25 | 90 | 17 | 0.2 | 490 |
| 6.5 | 0.5 | 0 | 15 | 0.5 | 90 | 18 | 0.5 | 490 |
| 8 | 1 | 0 | 18 | 1.0 | 90 | 22 | 1.0 | 490 |
| 18 | 2.7 | 0 | 23 | 1.6 | 90 | 25 | 1.5 | 490 |

The cohesive elements with zero thickness were inserted between the aluminum and polymer layers to estimate the bonding surface separation process; the Quads damage model is used for the bonding layer; more information is shown in the Table 6.

**Table 6.** Parameters of Quads damage model.

| Parameters | | Unit | |
|---|---|---|---|
| **Mechanical properties** | | | |
| Density | $\rho_0$ | (g/cm$^3$) | 1.1 |
| Stiffness matrix in normal direction | $K_n$ | (GPa) | 1.38 |
| Stiffness matrix in first shear direction | $K_s$ | (GPa) | 1.38 |
| Stiffness matrix in second shear direction | $K_t$ | (GPa) | 1.38 |
| **Quads damage model** | | | |
| Nominal stress in the pure normal mode | $\sigma_n$ | (MPa) | 80 |
| Nominal stress in the first shear direction | $\sigma_s$ | (MPa) | 40 |
| Nominal stress in the second shear direction | $\sigma_t$ | (MPa) | 40 |
| Displacement at failure | | | 0.5 |

## 3. Results and Discussion

### 3.1. Effect of Polyurea Coating on Energy Absorption

The impact resistances of the pure aluminum plate at 500 m/s, 1000 m/s, and 1400 m/s were analyzed. Figure 4 left displays the residual velocities of the projectile after hitting the target plate. The projectile velocity dropped from 466 m/s to 0 m/s and the aluminum plate was not penetrated. When the impact velocity increased to 862 m/s, 1027 m/s, and 1359 m/s, the impact velocity was reduced by 203 m/s, 182 m/s, and 190 m/s, respectively. This showed that the projectile velocity was attenuated at approximately 200 m/s and was almost unaffected by the impact velocity.

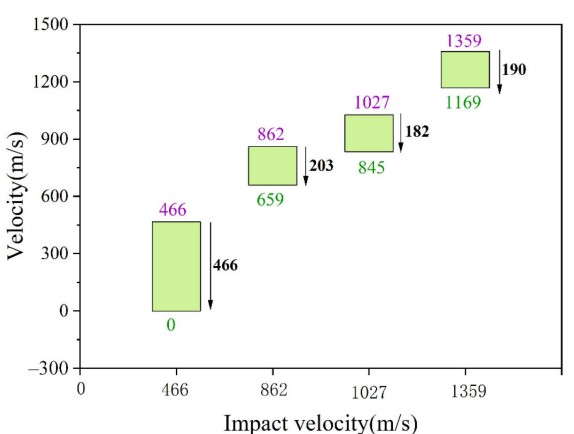
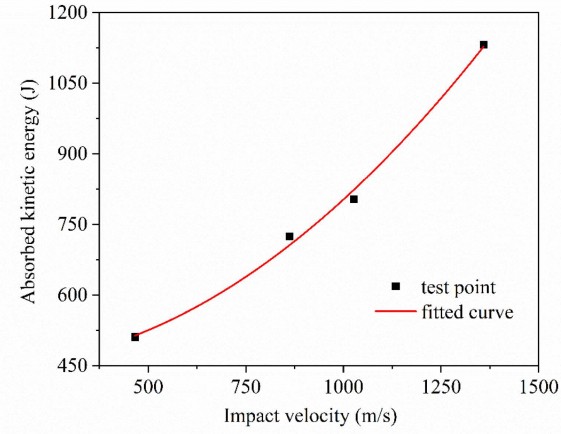

**Figure 4.** Velocity and kinetic energy attenuation in 10Al configuration.

However, the kinetic energy attenuation of the projectile (energy absorbed by the plate) became serious with the impact velocity increases. The phenomenon might have

been caused by the aluminum plate displaying a remarkable strain rate strengthening effect when subjected to a high-velocity impact. As shown in Figure 4 right, the absorbed kinetic energy increased in a nonlinear fashion with the initial velocity increase. The energy-absorbing performance of the aluminum plate gradually increased with the increase in the projectile velocity.

Two configurations, namely, double coating (1P/10Al/1P) and back coating (10Al/2P) were introduced to explore how the relative position of the polyurea affected the impact resistance (Figure 5). Both the double and back coatings were penetrated at the 552 m/s and 567 m/s impact velocities, and the velocities were reduced by 364 m/s and 398 m/s, respectively. Comparing the double coating and the back coating, the velocities dropped by 219 m/s and 246 m/s for the 1024 m/s and 849 m/s impact conditions, respectively (Figure 5 left). The results indicated that both the back and double polyurea coatings improved the impact resistance of the aluminum plate. It was difficult to state that the back coating performed better than the double coating in terms of the projectile velocity attenuation because the impact velocity at the back coating was lower than the double coating experimental condition.

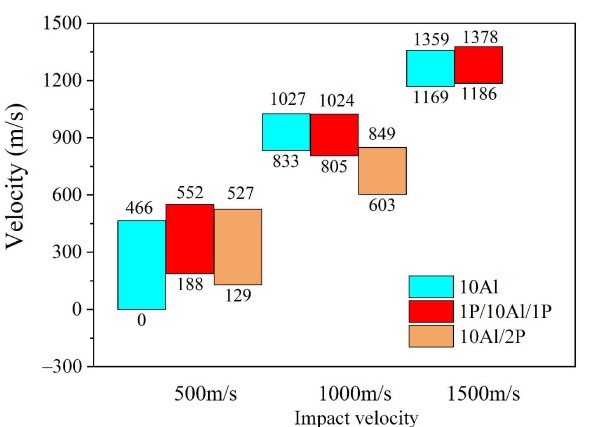 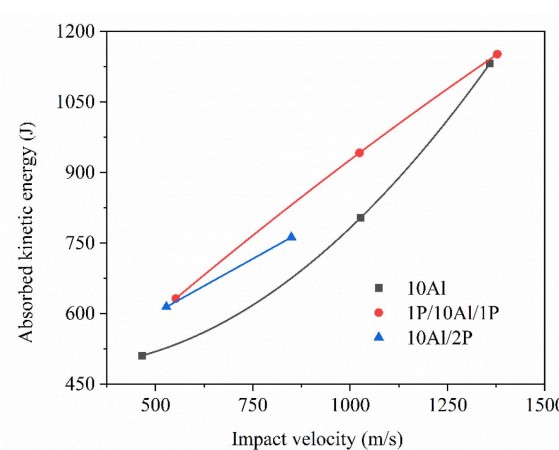

**Figure 5.** Velocity and kinetic energy attenuation in the 10Al, 1P/10Al/1P, and 10Al/2P configurations.

The velocity and kinetic energy attenuation with 2 mm, 4 mm, and 6 mm coating thicknesses are shown in Figure 6. For the same impact condition, the projectile velocity decay was obvious for the thicker coating. For instance, the velocities decayed to 398 m/s (10Al/2P) and 421 m/s (10Al/6P), corresponding to the 527 m/s and 529 m/s impact velocities (Figure 6 left).

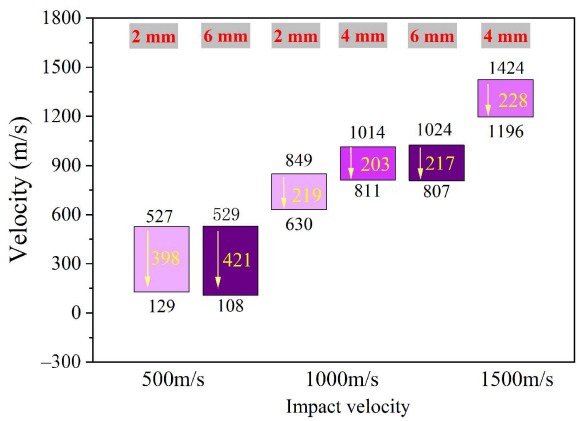 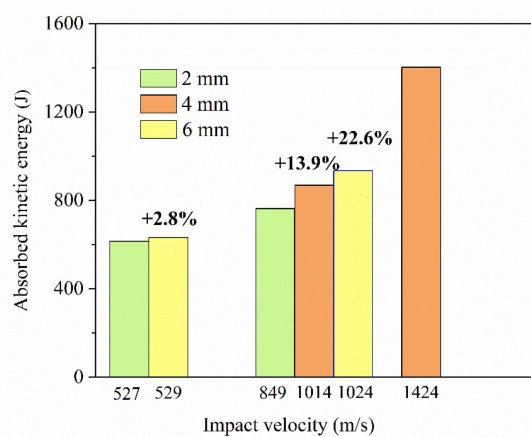

**Figure 6.** Velocity and kinetic energy attenuation in 10Al/2P, 10Al/4P, and 10Al/6P configurations.

As shown in Figure 6 right, the composite plate absorbed more impact energy with the increase in the coating thickness. For example, when the composite plate was subjected to 1000 m/s impact loading, the absorbed kinetic energies increased by 13.9% (10Al/4P)

and 22.6% (10Al/6P) compared to the 2 mm coating thickness configuration. Moreover, the impact velocity had a great effect on the kinetic energy decay process. When the composite plate was subjected to a lower velocity impact (500 m/s), the absorbed kinetic energy was simply increased by 2.8% (10Al/6P) compared with 10Al/2P.

### 3.2. Dynamic Failure of Coating Plate

Figure 7 displays the strain field evolution along the back face of the 10Al plate. When the projectile hit the aluminum plate at 1027 m/s, the aluminum plate was penetrated at 67 μs and the projectile flew away from the plate. The plastic deformation of the aluminum was formed around the hole. The deformation area expanded with the increase in the impact velocity to 1359 m/s. Moreover, obvious radial strain bands appeared around the bullet hole, and the average strain exceeded 0.005 at the 1359 m/s impact. The radial deformation was caused because the circumferential tensile stress exceeded the material tensile strength with the strong impact loading. The aluminum plate absorbed more kinetic energy with the large plastic deformation for the high-speed projectile impacts.

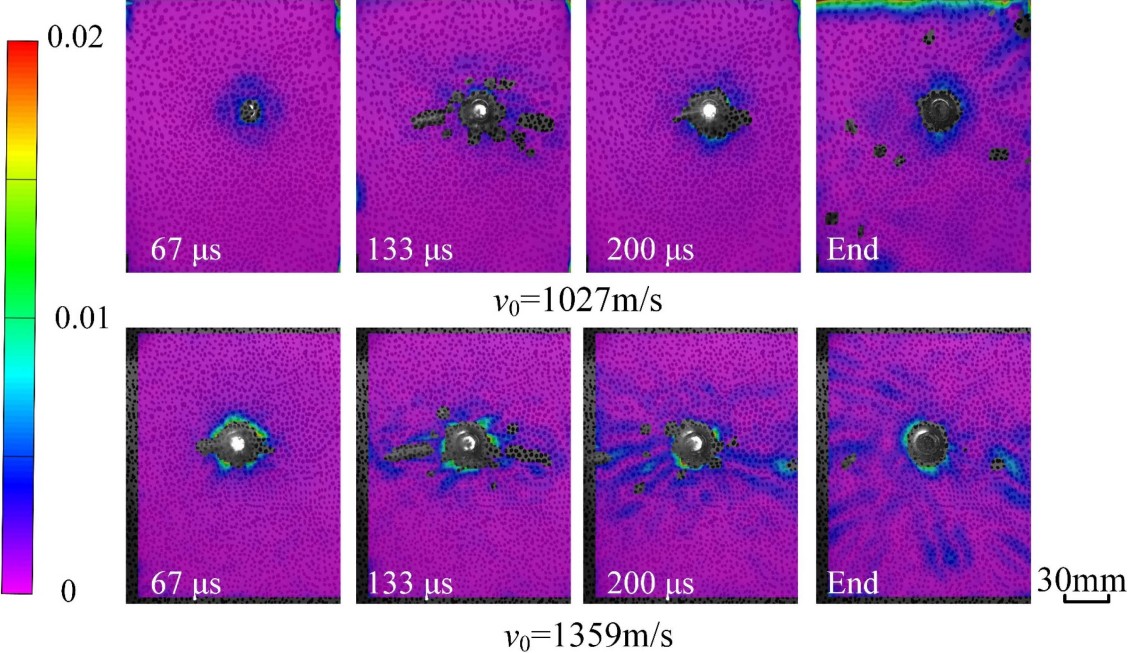

**Figure 7.** Strain field of 10Al with 1028 m/s and 1359 m/s impacts.

The dynamic responses of the polyurea coatings in the 1P/10Al/1P and 10Al/2P configurations are shown in Figure 8. Two frames, including the starting and ending deformation were extracted. For the lower velocity impact (500 m/s), the dynamic response processes of the 1P/10Al/1P and 10Al/2P plates were similar, and the perforation damage and the plastic deformation were obvious. The radial strain band was not significant in the polyurea coating because the deformation occurred in the polyurea elastomer with the circumferential tensile stress. When the impact velocity increased to 1378 m/s, the radial strain band appeared and spread outward to the boundary in the 1P/10Al/1P plate. The results showed that the radial deformation was easily formed with the higher speed impact, and the increase in the coating thickness was helpful for reducing the radial deformation.

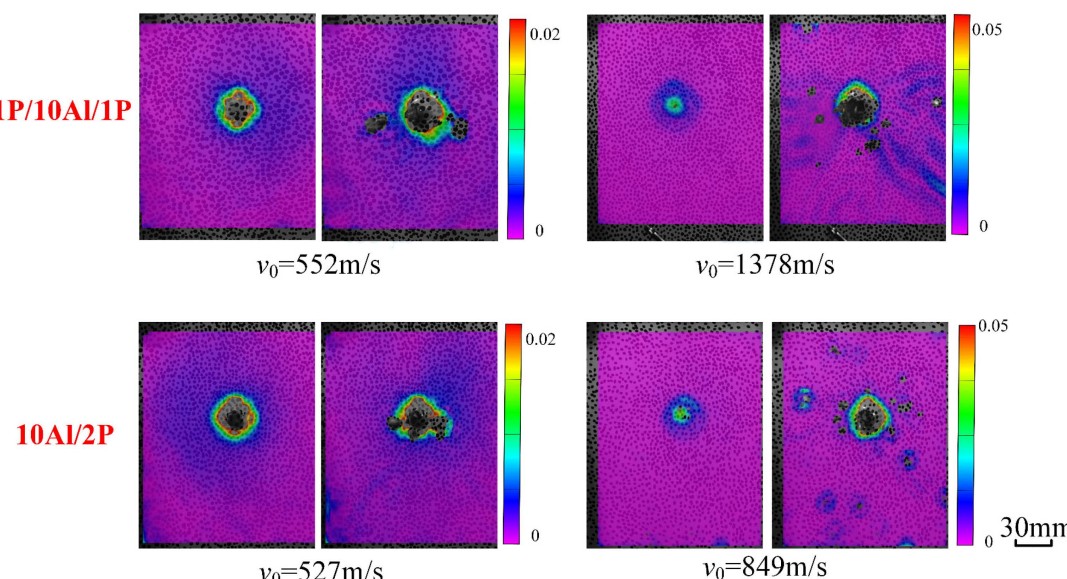

**Figure 8.** Strain field of 1P/10Al/1P under 552 m/s, 1378 m/s, impacting, and strain field of 10Al/2P under 527 m/s, 849 m/s, impacting.

The penetration and deformation processes of the 10Al/2P, 10Al/4P, and 10Al/6P plates at different impact conditions are displayed in Figure 9. The stress concentration field appeared around the hole with a low-velocity impact (500 m/s). The damage around the hole became serious with the impact velocity increasing. The fragments peeled off the composite board and flew away with the projectile. For the same impact conditions, the addition of the coating thickness was an effective way to absorb more impact kinetic energy with the plastic deformation of the polyurea materials.

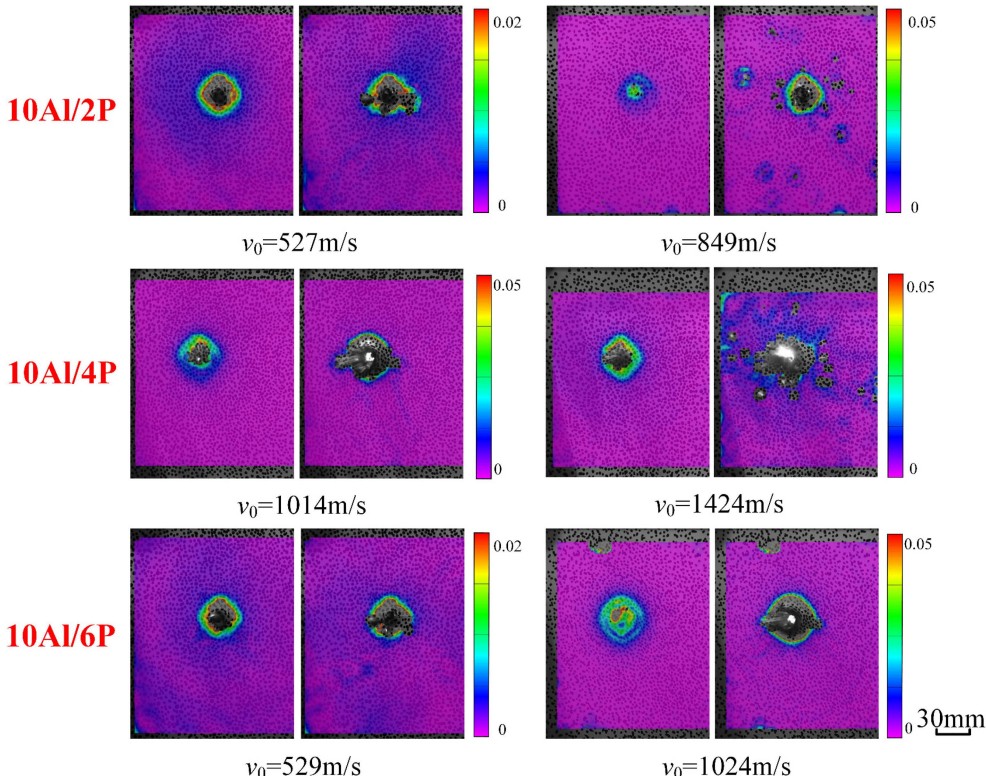

**Figure 9.** Strain field of 10Al/2P, 10Al/4P, 10Al/6P for 527 m/s and 849 m/s, 1014 m/s and 1424 m/s, 529 m/s and 1024 m/s impacts.

### 3.3. Numerical Results

The numerical modeling was conducted for different impact conditions by ABAQUS/ Explicit. The penetration process of the projectile to 10Al/2P composite plate at 849 m/s impact velocity and the penetration process of fragmentation to 10Al/6P composite plate at 1024 m/s impact velocity were simulated. As shown in Figure 10, the central deformation zone originates from the impact position and spreads out in concentric circles after impacting. The propagation process of strain fields is similar in experimental and simulation results for 10Al/2P and 10Al/6P samples, and the strain value in the experimental results is greater than it is from the simulation results. The difference is mainly caused by the digital image correlation (DIC) methods, the displacement of the partial failure zone is adopted when calculating strain in the unfailed region.

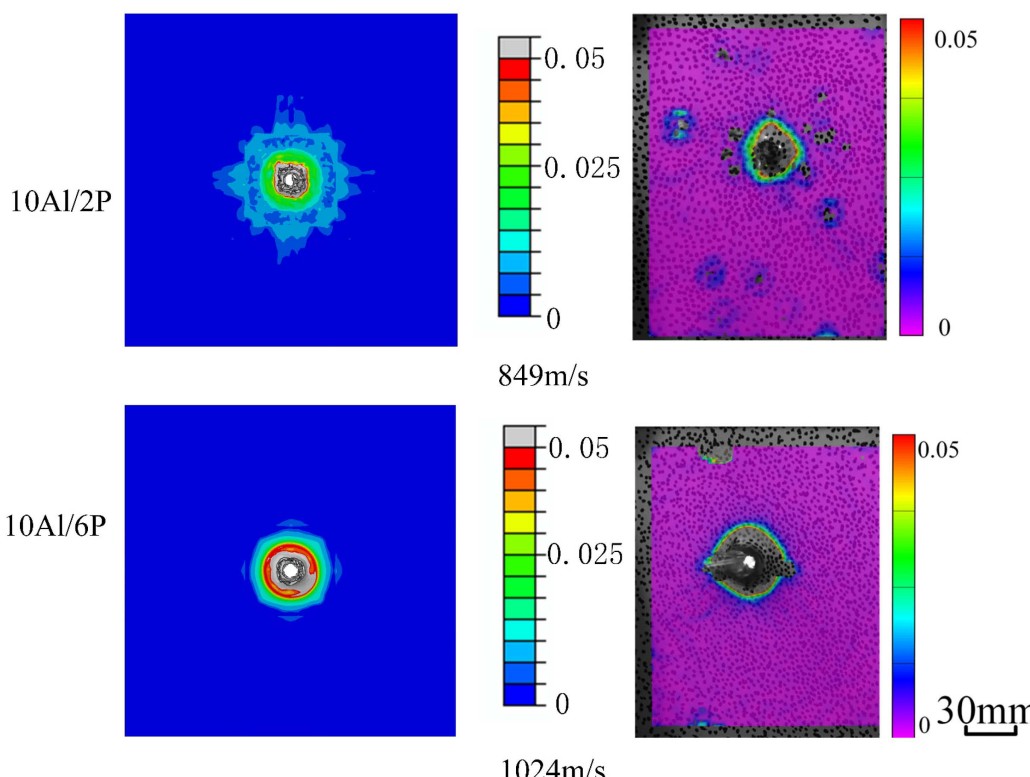

**Figure 10.** Strain field from experiments and simulations (1P/10Al/1P under 552 m/s, 1378 m/s; 10Al/2P under 527 m/s, 849 m/s.

The Von Mises stress fields of the 10Al plate for different impact velocities are displayed in Figure 11. The projectile's plastic deformation became serious with the increase in impact velocity, and the stress in the front area exceeded 0.8 GPa. The interactive force between the projectile and the plate increased significantly with the increase in the impact velocity. Ductile crater propagation was observed in the plate, and the ductility qualities of the aluminum became significant with the high-velocity impacts. The concentrated stress that was produced around the perforation hole was lower than 0.7 GPa, and the stress significantly increased by the reflected stress wave loading in the free boundary. Moreover, the perforation diameter increased with the loading velocity, and the petalling and petal bending was obviously with the 1359 m/s impacts. The results indicated that the shear failure played an important role in the aluminum plate perforation process with high-speed impacts.

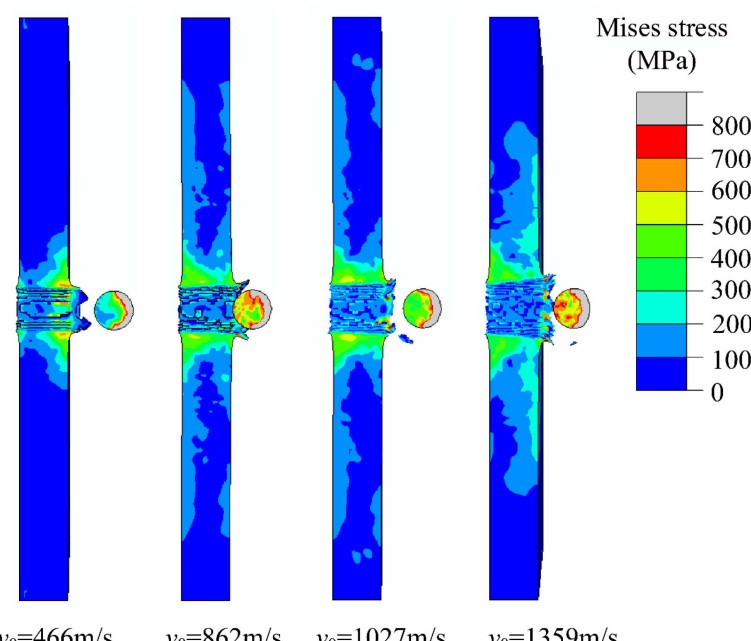

$v_0$=466m/s    $v_0$=862m/s    $v_0$=1027m/s    $v_0$=1359m/s

**Figure 11.** The stress fields of the 10Al plate for different impact velocities.

The kinetic energy absorption of the 10Al plate for different impact velocities in the numerical results was in good agreement with the experimental data (as seen in Figure 12). At lower velocity impacts, the kinetic energy from the numerical results was smaller than that of the experiment results, and this phenomenon was caused by the element deletion options in the simulation model. As the impact velocity increased, the absorbed kinetic energy was less than that of the numerical results because the temperatures during high-speed crushing was ignored in the simulation.

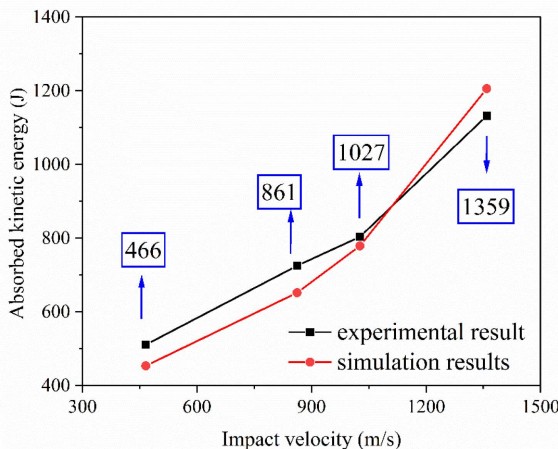

**Figure 12.** The kinetic energy absorption of the 10Al plate with different impact velocities.

As presented in Figure 13, the dynamic response was different in the front and back polyurea layers. The fracturing was combined with compression and shearing in the front polyurea layer, and the tensile failure played a dominant role in the damage to the back layer. The plastic deformation along the back layer was greater than that in the front polyurea layer because the polyurea had high elongation. The stress of the polyurea coatings was lower than that in the aluminum layer and absorbed the impact energy absorption through large deformation. For the polyurea elastomer located near the back face (10Al/2P), the plastic deformation and tearing happened only at the end of the penetration process.

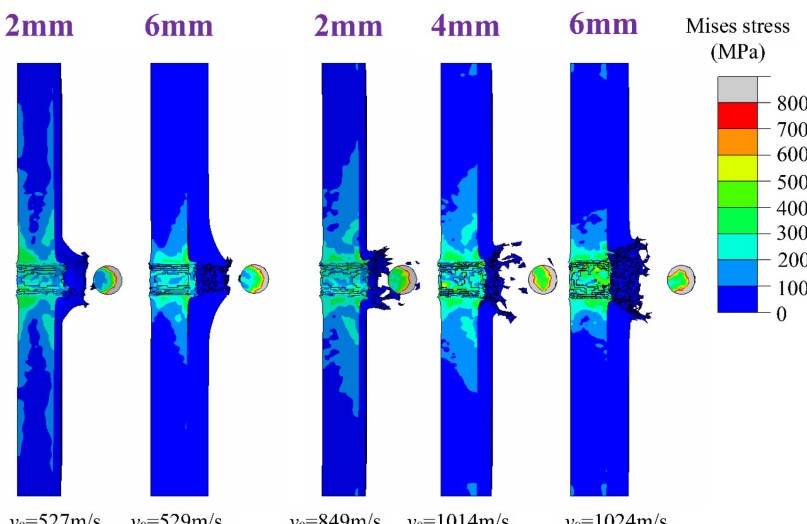

**Figure 13.** The stress fields of the 1P/10Al/1P and 10Al/2P plates for different impact velocities.

The Von Mises stress fields of polyurea coatings with different thicknesses are displayed in Figure 14. The tearing failure and the petalling penetration became marked by the increase in layer thickness. The tearing failure was noteworthy, especially in the high-speed impact condition, and the damaged area decreased with the increase in coating thickness for the same impact conditions.

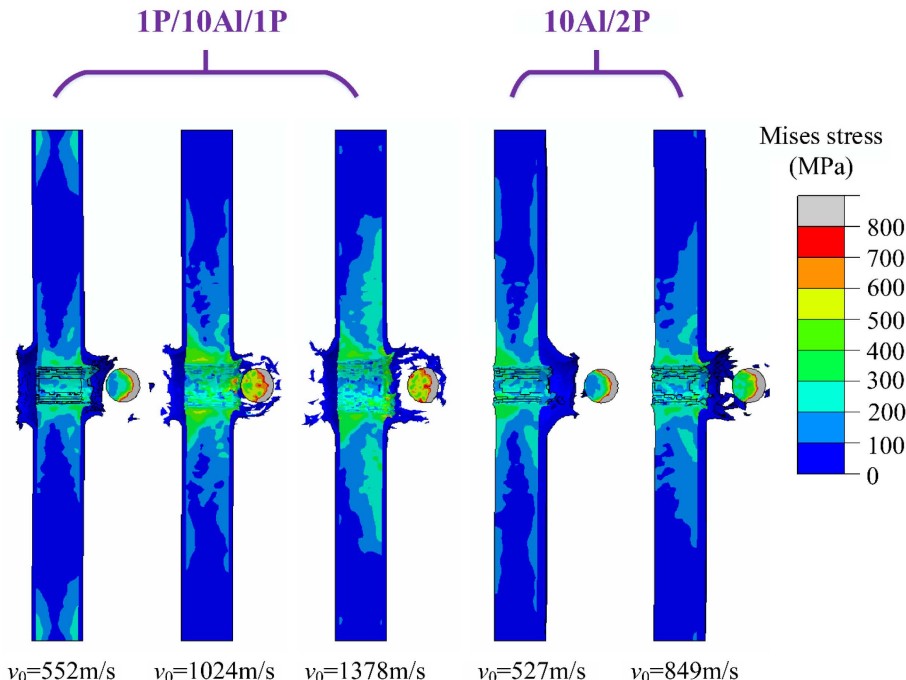

**Figure 14.** The stress fields of polyurea layer plates with different thicknesses for different impact velocities.

For removing the absorbed kinetic of energy disturbance from smaller coating thickness and impact velocity deviation, the 6P/10Al, 3P/10Al/3P, and 10Al/6P under 500 m/s, 1000 m/s, 1500 m/s are introduced to compare the impact property of the polymer with different laying mode (front coating/back coating/double coating). As shown in the Figure 15, the impact property of three composite plates is similar at the middle-low impact velocity range, and the performance of back coating (10Al/6P) is superior to others.

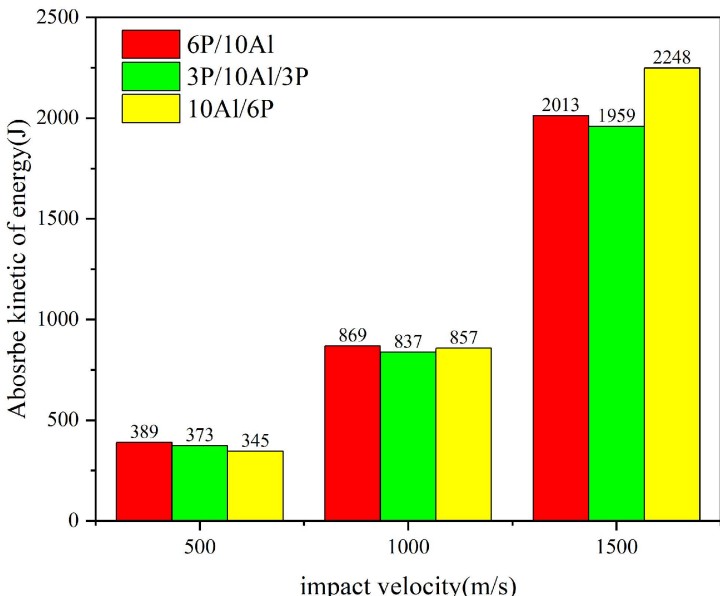

**Figure 15.** Kinetic energy in 6P/10Al, 3P/10Al/3P, and 10Al/6P configurations.

It has shown that the impact resistance property is similar when the impact velocity is small, and the front (or double-sided) coating has a smaller advantage in energy-absorbing performance than the back coating. When the impact velocity is between 1000–1500 m/s, the back coating has a better energy-absorbing performance than others.

## 4. Failure Mechanisms

### 4.1. Damage Appearance of Composite Layered Plate

The morphological characteristics of the aluminum plate (Al10) are displayed in Figure 16. The projectile penetrating radius increased with the impact velocity increase, and larger deformation was obviously produced around the bullet hole of the aluminum plate. The aluminum plates absorbed more kinetic energy by undergoing plastic deformation before they broke. Moreover, the impact pressure formed in the crushing failure along the perforation wall, and the bore diameter for the impact surface was greater than that for the back surface. The prominent deformation around the bore in the back face was not obvious when the back face was subject to higher velocity impacts because the adiabatic shear failure played a dominant role in the perforation process.

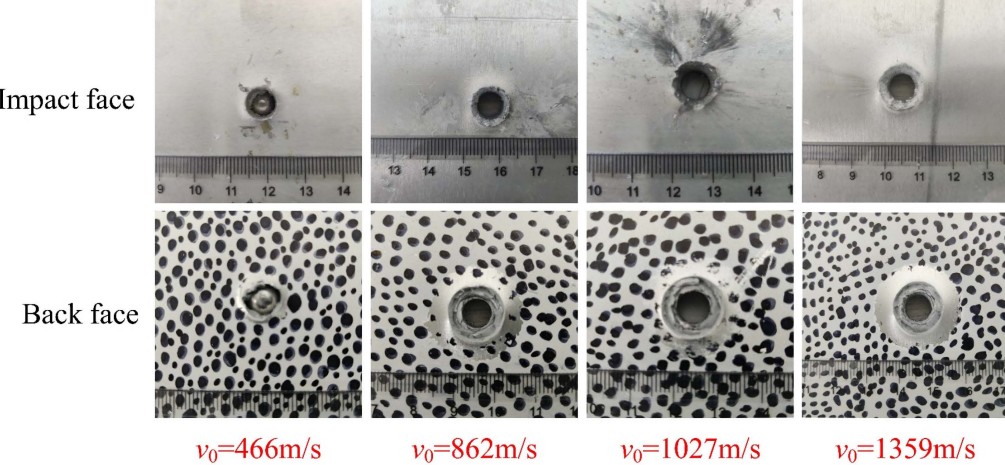

$v_0$=466m/s          $v_0$=862m/s          $v_0$=1027m/s          $v_0$=1359m/s

**Figure 16.** Damage appearance of 10Al for different impact conditions.

The de-bonded area and the damage were observable around the hole in the 1P/10AL/1P plate (Figure 17). In the impact face, the failure mode of the polyurea layer was similar at different impact speeds, and the slip damage boundary of the cylindrical perforation was produced by plugging the perforation. The swelling and cracking of the polyurea became serious with the impact velocity decrease because the deformation area of the aluminum plate extended with the increase in the penetrating time.

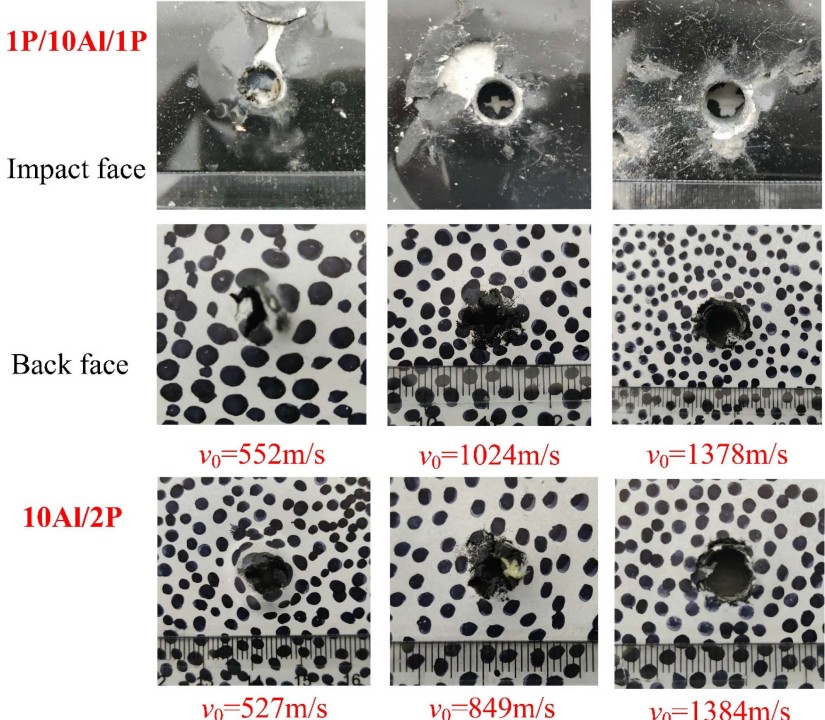

**Figure 17.** Damage appearance of 1P/10Al/1P and 10Al/2P with different impact conditions.

The failure characteristics of the front and back coatings were different. The non-recoverable deformation of the spray polyurea was produced at 522 m/s impacts, and the curled and conglutinated phenomenon was obvious around the hole. The coking polyurea particles were produced in the black coating with high-speed impacts because of the transient contact temperature rise in the process of interaction.

The perforated damage for different coating thicknesses is displayed in Figure 18. The tearing area decreased with the coating thickness increasing under the same impact conditions. In addition, the black points were obvious around the central damage zone when the zone was subject to higher velocity impacts. This phenomenon might have arisen from the high local temperature and material softening with high-speed projectile impacts.

### 4.2. Effect of Polyurea Coating on Energy Absorption

An HS camera technique was used to observe the penetration process. The destruction process on the back surface of the 10Al/2P plate with 527 m/s loading is displayed in Figure 19. After the projectile went through the specimen at 67 µs, the damage to the polyurea layer was concentrated at the impact location. When the projectile flew away from the back face at 200 µs, the sphere shape remained in an intact state and the conspicuous damage was not formed because the projectile had high strength. The force of the collision spewed large amounts of debris from the composite panel following the projectile after 200 µs.

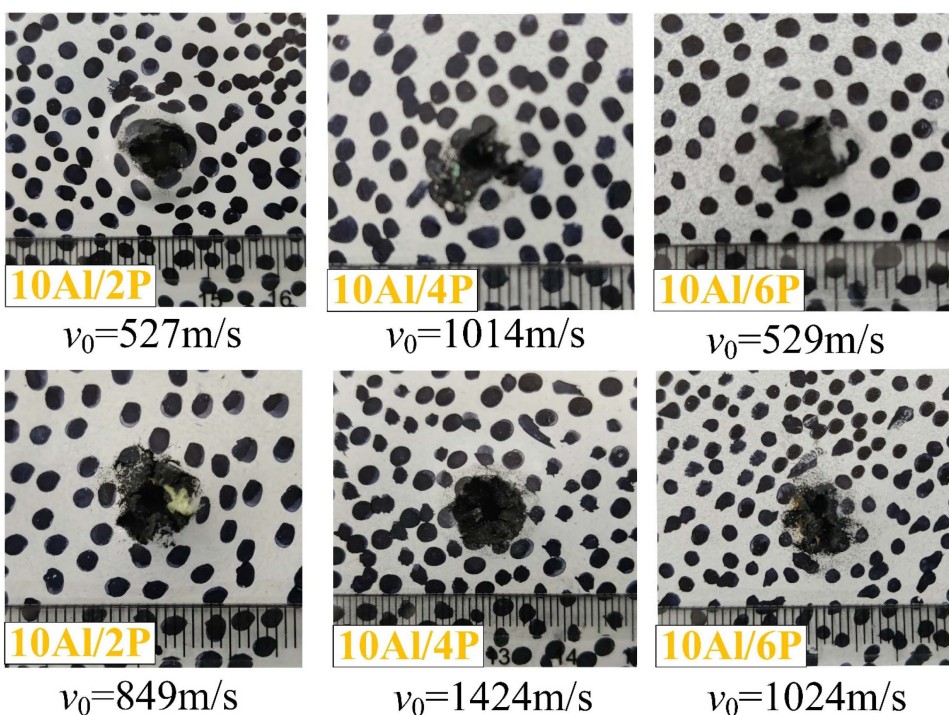

**Figure 18.** Damage appearance of the back face in the different polyurea layer thicknesses for different impact conditions.

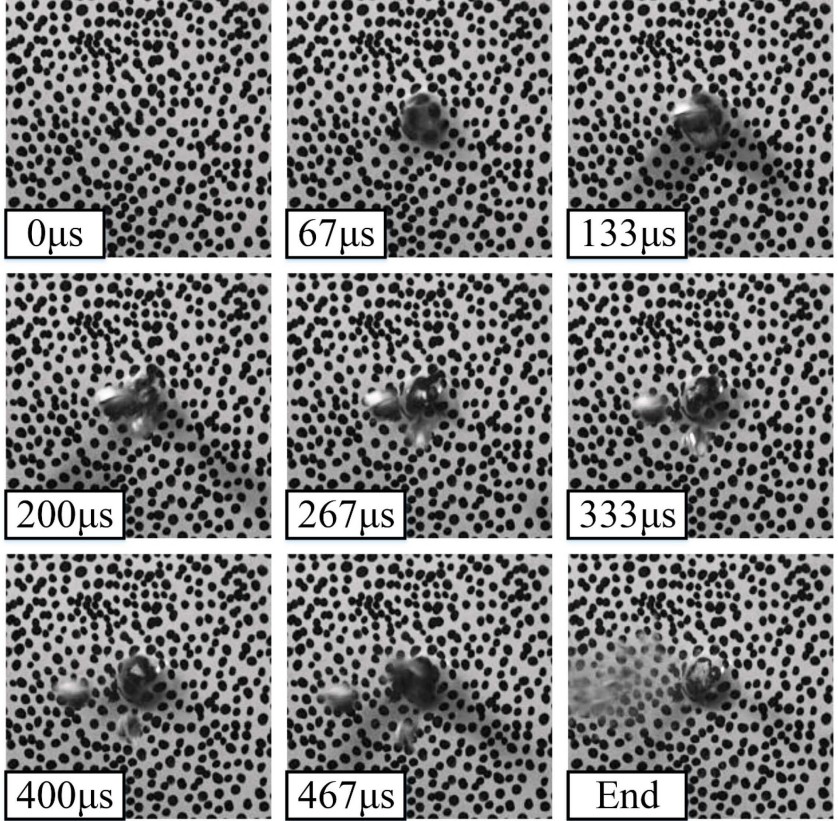

**Figure 19.** Penetration process of 10Al/2P plate with 527 m/s loading.

To understand the internal failure of the composite panel structure, the fractures and the microstructure bonding failure were observed using computed tomography (CT). The X-ray 2D imaging was tested in the Sanying Precision Instruments Co. Ltd in Tianjin,

China. The detection area was 120 mm × 12 mm and the resolution ratio was 50 μm by multiscaleVoxel-2000 industrial CT. Both the longitudinal section (along the impact direction) and the transverse section were extracted.

The composite structure included the top coating, top bonding interface, aluminum plate, lower bonding interface, and lower coating five parts. The test results from X-ray CT scans are shown in Figure 20. The wear and damage are clearly observed in the micrograph. When the projectile impact the front polyurea coating, the maximal stress exceeds the compressive strength limit of polyurea and partial polyurea coating peels off the aluminum sheet, because the bonding interface is separated by the reflected wave. When the projectile enters the aluminum plate, the shear failure causes the shear band to arise around the hole and become observable along the back bonding face. In the back polyurea coating, the separated circular area is smaller than the front bonding interface, because the polyurea's ductility and malleability increase with the temperature increase. The tensile fracture caused the petaloid cracking in the back polyurea coating.

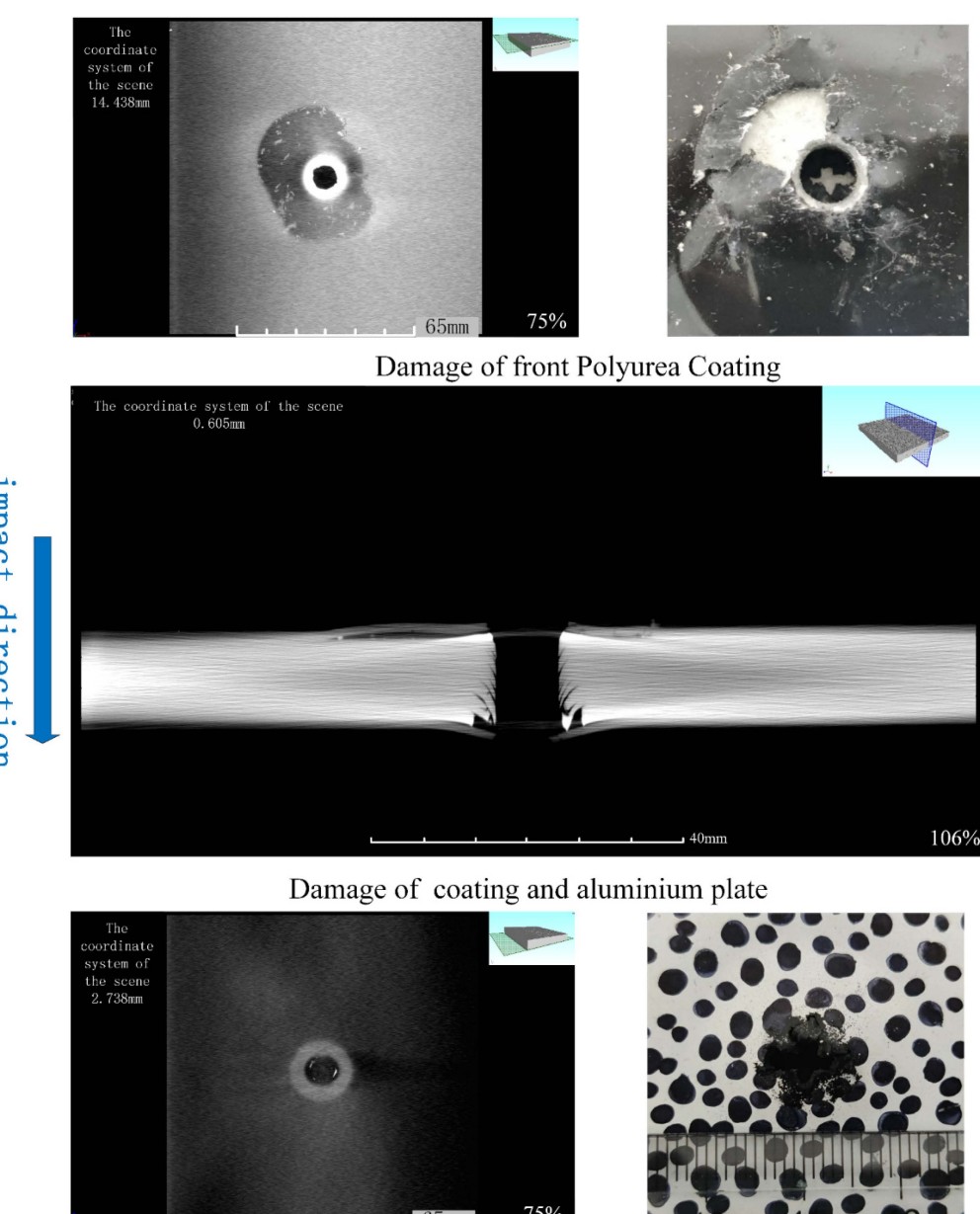

Figure 20. Microstructure fracture of 1P/10Al/1P by CT test.

## 5. Conclusions

The effects of the spraying thickness and relative position on response plates under dynamic loading are studied by experiment and simulation. Five experiments were used to study the penetration process of polyurea-coated 2024 aluminum plates with the impacts of tungsten spheres. DIC and CT methods were adopted to calculate the strain field and analyze the microscopic damage.

The results indicate the impact resistance performance is similar at slower impact velocity (500–1000 m/s), and the front (or double-side) coating has a smaller advantage. When the impact velocity rise to 1500m/s, the back coating has a better energy-absorbing performance. Under the same impacting conditions, adding coating thickness was an effective way to absorb more impacting kinetic energy with the plastic deformation of polyurea materials.

The kinetic energy absorption from numerical results was in good agreement with the experimental data. The fracturing was combined with compression and shearing in the front polyurea layer, and the tensile failure played a dominant role in the damage of to the back layer. The swelling and cracks of the polyurea were more serious with the decrease in the impact velocity because the deformation of the aluminum plate closed the bore increased with the increase in the penetrating time.

The inside fracture and the microstructure bonding failure were observed using CT. The separated phenomenon was serious in the front bonding surface with shear compression failure. The compressional deformation was produced during the tungsten ball movement into the aluminum plate, and the shear band was formed along the hole walls. In the back polyurea layer, the stripping area was smaller than the front bonding interface because the tensile fracture caused the petaloid cracking. Moreover, the damage boundary of the top polyurea coating was smooth with the punching failure, whereas the irregular damage boundary was generated in the low polyurea coating.

**Author Contributions:** Writing—original draft, C.H.; Writing—review & editing, Y.L.; Formal analysis, Y.Y.; Methodology, Q.C. All authors have read and agreed to the published version of the manuscript.

**Funding:** This research was funded by the by the Foundation of Hubei Key Laboratory of Blasting Engineering (BL2021-09); the Opening Project of the Shanxi Province Key Laboratory of Functional Nanocomposites, North University of China (NFCM202101); the China Postdoctoral Science Foundation (2021M702981); the Fundamental Research Program of Shanxi province (20210302124197).

**Acknowledgments:** This paper was supported by the Hubei Key Laboratory of Blasting Engineering and Shanxi Province Key Laboratory of Functional Nanocomposites.

**Conflicts of Interest:** The funders had no role in the design of the study; in the collection, analyses, or interpretation of data; in the writing of the manuscript; or in the decision to publish the results.

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
