# Peer review of "Experimental and Numerical Investigation of Ballistic Resistance of Polyurea-Coated Aluminum Plates under Projectile Impacts"

_crystals, doi:10.3390/cryst13071039_

Round 1
Reviewer 1 Report
The article deals with numerical modeling of the impact of tungsten ball impact on the process of penetration of aluminum plates coated by Polyurea Coatings.
Introduction sufficiently describes the current state of knowledge with reference to the literature and justifies the research topic.
Materials and Experimental are described, but give more details about "Special spraying equipment and a spray gun were used"
The test results are clearly described at the experimental and numerical simulation stage, which can illustrate the properties of the tested materials. However, this cannot explain the occurring phenomena and mechanisms of wear and damage. The degree of detail of material tests is very low. I believe that material research, in particular structural research, morphology research should be at a much higher level of detail. Meanwhile, the authors present the morphology of the material at the macro level. In many cases there is no scale at all in the pictures presented. The claim that something concerns the "microstructure interface" is not justified, because the presented results of morphology research are not on a micro scale. I believe that the subject of the article concerns the study of material destruction, so such microstructural studies are indicated here. Additionally, little information is given regarding the characteristics of the protective layer itself. Its structure or some other characteristics allowing to characterize it should also be given. I believe that such research should be supplemented to give a more scientific character to this article.
Otherwise, the authors should modify the title of the article, clearly indicating in its content that it concerns numerical modeling and the content of the article should focus on the scientific aspects of modeling itself.
Author Response
The supplementary instruction and format errors have been added or modified according to suggestions, and more detailed information, Please see the attachment.

Reviewer 2 Report
The work presented investigates the effects of polyurea coatings on the response of aluminum panels subjected to hard-body impacts. While the topic is interesting, there are several comments that the reviewer recommends be addressed to improve the scientific rigor and clarity of the study.
1) Lines 96-97: "several metallic specimens were coated with 1–6 mm (back side) and 1 mm (double sides) thickness polyurea layers" - From Table 1 and the rest of the manuscript, it appears that the backcoated samples were coated with only 2, 4, or 6 mm of coating. Therefore, it may be inappropriate to mention the backcoat thickness in the range of 1-6 mm.
2) Lines 120-121: "The area of interest (AOI) size was 200 × 200 mm with approximately 320 × 320 pixels (as seen in Figure 2)." - compare with lines 249-250: "The strain contour area was 150 mm × 200 mm and it consisted of the original image and calculated picture with 20% opacity." - What was the size of the monitored area 200 mm x 200 mm or 150 mm x 200 mm?
3) Lines 189-191: "The material parameters were obtained from different stress-strain curves of polyurea at different strain rates in the literature [7] (Table 3)." - Table 3 shows only basic material properties for a single strain rate. Where are the others? How was the strain rate effect implemented?
4) Figure 4 - What do the values on the x-axis of the top graph represent?
5) The authors should state how the absorbed kinetic energy was evaluated. If the weight reduction of the projectile was not measured and the absorbed energy is basically just the difference of the squared velocities before and after perforation, then I do not understand why both velocity reduction and energy absorption are discussed. Either use only one of these parameters or explain in detail why both parameters need to be mentioned and discussed.
6) Lines 225-227: "As shown in Figure 5b, the 1P/10Al/1P construction absorbed more of the kinetic energy of the projectile impact compared with 10Al/2P at the same loading condition, especially for the 500 m/s impact condition." - I don't think this statement is provable. The absorbed energy values of Sample 1P/10Al/1P and Sample 10Al/2P at speeds of about 500 m/s are practically the same in Figure 5.
7) Figures 7, 8, and 9 - Would it be possible to output the strain fields from numerical models and compare them with the recorded ones?
8) Lines 420-421: "The aluminum plate with double-sided coating had a better energy absorbing performance than that with back coating." I am struggling to find enough data to support this conclusion. Lines 221-224 reads "It was difficult to state that the back coating performed better than the double coating in terms of the fragment velocity attenuation because the initial velocity at the back coating was lower than the double coating experimental condition." The energies absorbed by a single coated specimen and a double coated specimen are very similar in Figures 5 and 13. If you cannot draw a conclusion based on measured values, you should not simply square those values, subtract one from the other, multiply by constants, and then make a statement.
9) Line 130: "Five groups of specimens were performed in this study." - This is just one example of many language deficiencies. The manuscript should be proofread to improve the level of English.
10) Authors use too many different words to refer to the same thing, which can be confusing on first reading. See, for example, tungsten ball, projectile, fragment, and bullet. I would recommend choosing one word and sticking to it throughout the manuscript. I would not recommend calling the projectiles fragments. They can be used to represent the fragment of the warhead, but they are still not fragments.
11) A similar case is the name of the velocity measured between the gun and the specimen. Sometimes, it is called initial velocity, impact velocity, and sometimes even initial impact velocity. If measured near the barrel, it should be called initial or muzzle velocity. If measured near the specimen, it may be called impact velocity.
Author Response

(The authors gave the same response as above.)

Round 2
Reviewer 1 Report
The authors supplemented the content of the article. Added scale in photos. Wording about “micro” has been removed as studies were not presented at this level of detail. The title of the article has also been changed. The current title better describes the content of the article.